# Self-Healing Chitosan Hydrogels: Preparation and Rheological Characterization

**DOI:** 10.3390/polym14132570

**Published:** 2022-06-24

**Authors:** Anda Mihaela Craciun, Simona Morariu, Luminita Marin

**Affiliations:** “Petru Poni” Institute of Macromolecular Chemistry, Gr. Ghica Voda Alley, 41A, 700487 Iasi, Romania; smorariu@icmpp.ro

**Keywords:** chitosan, pyridoxal 5-phosphate, hydrogels, self-healing, rheology

## Abstract

The paper aims at the preparation of chitosan self-healing hydrogels, designed as carriers for local drug delivery by parenteral administration. To this aim, 30 hydrogels were prepared using chitosan and pyridoxal 5-phosphate (P5P), the active form of vitamin B6 as precursors, by varying the ratio of glucosamine units and aldehyde on the one hand and the water content on the other hand. The driving forces of hydrogelation were investigated by nuclear magnetic resonance (NMR), Fourier-transform infrared spectroscopy (FTIR), X-ray diffraction, and polarized light microscopy (POM) measurements. NMR technique was also used to investigate the stability of hydrogels over time, and their morphological particularities were assessed by scanning electron microscopy (SEM). Degradability of the hydrogels was studied in media of four different pH, and preliminary self-healing ability was visually established by injection through a syringe needle. In-depth rheological investigation was conducted in order to monitor the storage and loss moduli, linear viscoelastic regime, and structural recovery capacity. It was concluded that chitosan crosslinking with pyridoxal 5-phosphate is a suitable route to reach self-healing hydrogels with a good balance of mechanical properties/structural recovery, good stability over time, and degradability controlled by pH.

## 1. Introduction

Hydrogels are three-dimensional polymeric networks with high water content, which found a great variety of applications in biomedicine, agriculture, or environmental protection. Recent studies revealed self-healing (SH) hydrogels as a new type of soft matter with a unique ability to recover the structure and function after applying an external stimulus, such as an injection through a needle [1]. Thus, the SH hydrogels are valuable materials, which offer the advantage of simple administration by injection at a certain site of the body, to fill a cavity with an irregular shape or to form a coating on a surface in tissue engineering or wound healing. Moreover, by loading with bioactive compounds, they transform into excellent vehicles for drug delivery systems to be applied for local action, in tumors or on wounds [2,3]. Many efforts were directed toward the development of SH hydrogels. Rationally speaking, the SH hydrogels can be prepared via non-covalent or reversible covalent bonds that break under external stress and reform when the stress is removed. Thus, the SH mechanism is grounded on the dynamic equilibrium between dissociation and recombination of various interactions, which allows the hydrogel to heal damages and reform shapes. Studies on SH hydrogels demonstrated that while physical interactions lead to a fragile and rapid dynamic equilibrium, the dynamic covalent bonds lead to a stable and slow one [4]. Common reversible dynamic bonds used for building SH hydrogels include imine, boronate, oxime, disulfide, and acylhydrazone bonds, or Diels–Alder reactions, whereas non-covalent interaction includes hydrogen bonds, or ionic, host–guest, and hydrophobic interactions [5]. SH hydrogels are often prepared using synthetic polymers such as polyvinyl alcohol, poly(ethylene oxide), poly(N-isopropylacrylamide), polyacrylic acid, polystearyl methacrylate, poly (N,N-dimethylacrylamide) or polyoxyethylene acrylate, which present the main disadvantage of being non-biodegradable, limiting thus the in vivo application [1,3,5]. To overcome this drawback, natural originating polymers are desirable alternatives, chitosan being a promising one [6]. Chitosan is a natural originating polymer with good biocompatibility and biodegradability, which is already broadly used in pharmaceutical and medical fields [7]. From the chemical point of view, chitosan is a polyamine, and its main reaction path is the formation of imine bonds, which are well-recognized reversible bonds with great reaction speed under mild conditions [8,9,10]. The literature survey reveals chitosan self-healing hydrogels prepared via imine bonds with a large variety of crosslinkers: glyoxal [11], glutaraldehyde [12], polyethylene glycol [13], Pluronic-F127 [14], sodium alginate [15], konjac glucomannan [16], and xanthan gum [17]. Nevertheless, the glyoxal and glutaraldehyde were demonstrated as being toxic (mutagenic and neurotoxic) [18,19], while the synthetic polymer crosslinkers are non-biodegradable.

Our studies in the field of chitosan hydrogels demonstrated a new crosslinking route of chitosan with monoaldehydes, based on the formation of imine units and self-assembling into clusters with the role of crosslinking nodes [20,21,22,23,24,25,26]. The method is promising, giving access to a large variety of multifunctional hydrogels, whose properties can be tuned by a proper choice of the monoaldehyde crosslinker. Moreover, the aldehydes exist in a large variety in nature, having a high potential to be biocompatible. Such an aldehyde is pyridoxal 5-phosphate (P5P), the active form of vitamin B6, approved by the FDA for human usage as a food supplement, which proved preventive tumorigenesis activity [27] and synergistic effect with some antitumor drugs [28]. In this framework, the hydrogels prepared from chitosan and P5P keep the promise of active carriers for antitumor drug delivery in local therapy. The rational question is if these hydrogels based on the reversible imine bonds present a self-healing effect and can be administered at the tumor site by injection. To find the answer to this question, a large series of hydrogels based on chitosan and P5P were prepared, and the influence of their composition on the mechanical and thixotropic properties was investigated by rheological measurements. Moreover, their self-healing ability was verified by laboratory tests. It was demonstrated that hydrogels based on chitosan and P5P have self-healing ability and can be easily applied by injection.

## 2. Materials and Methods

### 2.1. Materials

Low molecular weight chitosan (Ch) (Mv = 169.9 kDa and DD = 83% established by viscosimetry and NMR methods [29]), pyridoxal 5-phosphate (≥98%), acetic acid (HAc) (≥ 99%), sodium acetate (≥ 99%), disodium hydrogen phosphate (for analysis), potassium dihydrogen phosphate (for analysis), sodium chloride (≥ 99%), potassium chloride (≥ 99%), and sodium hydroxide (≥ 99%) from Aldrich (Darmstadt, Germany) were used without further purification.

### 2.2. Synthesis

A large series of 30 hydrogels were prepared by mixing chitosan with pyridoxal 5-phosphate in an aqueous acetic acid solution, varying the molar ratio of glucosamine units of chitosan and P5P (NH_2_/CHO ratio) on the one hand, and the water content on the other hand (Table 1). Then, 2 mL solution of 3.985–0.455% P5P in 0.35% aqueous HAc was added under vortex shaking to a chitosan solution (3%, 1%, 0.6%, and 0.43%), at 55 °C (Table 1). Hydrogelation instantaneously occurred for the mixtures with high content of P5P and slower for those containing lower content of P5P. All samples were obtained starting from 60 mg chitosan dissolved in: 4 mL of water (**1**–**9**); 8 mL of water (**1d**–**5d**); 12 mL of water (**1t**–**4t**); or 16 mL of water (**1q**–**2q**) (Table 1).

The hydrogels were coded with numbers and letters to reflect their composition. Thus, the number indicates the molar ratio between NH_2_/CHO (from **1** to **9**), and the letter indicates the increase in the water content in hydrogel: no letter indicates the use of a 3% solution of chitosan; d: 1% (double water content in hydrogels); t: 0.6% (triple water content in hydrogels); q: 0.43% (four times higher water content in hydrogels), e.g., **2t** is the code for the hydrogel with a 2/1 ratio of NH_2_/CHO, triple diluted compared to the most concentrated hydrogel (3%).

### 2.3. Equipment and Methods

The solid hydrogel samples (xerogels) were prepared by hydrogel freezing in liquid nitrogen, followed by drying under reduced pressure (1.150 mbar) at 50 °C for 24 h, using a Labconco Free Zone Freeze Dry System equipment (Kansas, MO, USA).

NMR spectra were recorded on a Nuclear Magnetic Resonance (NMR) BRUKER Advance DRX 400 MHz spectrometer (Billerica, MA, USA), at room temperature, on hydrogel samples prepared in the NMR tubes using deuterated water.

FTIR spectra were recorded on Bruker Vertex 70 FTIR Spectrophotometer (Ettligen, Germany) using the ATR module on xerogel samples. OPUS 6.5 and OriginPro 8.5 software (Ettligen, Germany) were involved in the deconvolution of spectra in the 1700–1580 cm^−1^ region, using the curve fitting method with Lorentzian 50% and Gaussian 50%.

The supramolecular structure of the hydrogel was investigated with a diffractometer Benchtop Miniflex 600 Rigaku (Tokyo, Japan), from 5 to 40°, registered with 0.01 step and 3°/min speed, on the xerogel pellets, and with a Polarized Optical Microscope (Zeiss Axio Imager.A2m, camera Axiocam 208 cc (Wetzlar, Germany)), on thin xerogel slices.

Scanning electron microscope (SEM) was used to assess the morphology of the hydrogels by observing the xerogels with an SEM-EDAX–Quanta 200 (Eindhoven, Germany) using a field emission and voltage of 20 keV.

The hydrogels’ stability in media of different pH was investigated by gravimetric method, as follows. The hydrogel samples were weighed to contain 5 mg of dried substances (m_i_). Each sample was immersed in a 5 mL buffer solution of different pH at 37 °C for 9 days, and the buffer solution was refreshed every 3 days. After 9 days, the samples were taken out, washed three times with distilled water to remove salts residues, and then lyophilized. The obtained xerogels were weighted (m_x_), and the mass loss was calculated with the equation: mass loss = (m_i_ − m_x_)/m_i_ × 100, for each pH, at 9 days. The investigation was conducted in buffer media of different pH, as follows: pH = 5.6 (HAc/NaAc), pH = 6.8 (DHP, PDP, NaCl, and KCl), pH = 7.4 (DHP, PDP, NaCl, and KCl),) and pH = 8.0 ((DHP, PDP, NaCl, KCl, and NaOH).

The rheological investigations were carried out on an MCR302 Anton-Paar rheometer (Graz, Austria) with the plane–plane geometry (diameter of 50 mm). The rheometer is equipped with Peltier temperature control, and the solvent evaporation was prevented by using a trap cover (Malvern Instruments Ltd., Worcestershire, UK). The linear viscoelastic regime (LVR) for each sample was determined by an amplitude sweep test at a constant oscillation frequency (ω) of 10 rad s^−1^ in the shear stress (τ) range of 10^−3^–2 × 10^2^ Pa. The storage (G′) and the loss (G″) moduli values were determined by a frequency sweep test in the oscillatory frequency (ω) range of 10^−1^–10^2^ rad s^−1^ at a strain (γ) value from LVR. Finally, the structural recovery capacity of the investigated samples after applying three consecutive flow steps, 1–100–1%, at 10 rad s^−1^, was evaluated. The rheological measurements were performed at 37 °C in duplicate to estimate the variability of the results, using a fresh sample for each test. The errors were lower than 13%. For some strongly crosslinked samples, no rheological measurements could be performed due to their brittle properties.

## 3. Results and Discussions

### 3.1. Synthesis and Structural Characterization

A large series of 30 hydrogels was prepared by reacting chitosan with pyridoxal 5-phosphate (P5P) in different molar ratios between glucosamine units of chitosan and P5P aldehyde, and using different concentrations of the chitosan solutions, to reach hydrogels with different water content (Figure 1, Table 1). The hydrogel formation was determined by tube inversion test and by rheological experiments.

Structural and supramolecular characterization of the hydrogels revealed that hydrogelation took place due to the formation of imine units and the occurrence of physical forces between the chitosan chains and P5P.

The formation of the imine units was confirmed on the one hand through ^1^H-NMR spectra that displayed the occurrence of the specific chemical shift of imine proton around 9 ppm and on the other hand through FTIR spectra by the occurrence of specific vibration band of imine bond around 1602–1613 cm^−1^ (Figure 1a–d) [23,30]. It should be mentioned that ^1^H-NMR spectra also revealed the presence of aldehyde proton, in agreement with an equilibrium state of the imination reaction, shifted to the products when the functional amine units were in excess compared to aldehyde ones [20,22].

Further, the intermolecular forces were confirmed by the shifting of the maximum of the broad band characteristic for the vibration of the hydroxyl, amine, and H-bonds between them, from 3300 cm^−1^ in chitosan to 3600 cm^−1^ in hydrogels. This was associated with the modification of the H-bond environment, as well documented for other chitosan hydrogels [20,21,22,23,31]. The occurrence of intermolecular forces was also demonstrated by the X-ray diffraction of the hydrogels when compared to pristine chitosan. Chitosan displayed two broad overlapped diffraction bands from 8 to 14° and from 15 to 27°, corresponding to the inter-chain and intra-molecular distances, respectively, in line with the presence of ordered clusters characteristic for the semicrystalline nature of chitosan [32]. By comparison, the hydrogels showed only a broad diffraction band from 12 to 27°, indicating a larger polydispersity of intermolecular distances (Figure 1e). This is in accordance with the (i) weakening of the intermolecular forces between the chitosan chains caused by the grafting of the imine units and (ii) manifestation of new intermolecular forces, i.e., H-bonds prompted by the hydroxyl groups of phosphate units of P5P. These data indicate that chitosan hydrogelation was driven by the imination and physical forces induced by the newly formed imine units. On the other hand, POM images showed intense birefringence, the signature of an ordering degree (Figure 1f) [20,21,22,23,24,25,33]. Coupling X-ray and POM data, it can be envisaged that intermolecular forces directed a supramolecular arrangement of the imino-chitosan chains.

### 3.2. Morphology

The morphological appearance of the hydrogels was investigated by SEM analysis of the corresponding xerogels (Figure 2). All hydrogels displayed microporous morphology, with pore dimensions varying as a function of water content and crosslinking degree. Thus, as expected, the increase in the water content in hydrogels was accompanied by the increase in the pore diameter from around 10 μm to approx. 50 μm. Against the rule reported in the literature for the covalent crosslinked hydrogels, according to which the pore dimension increases as the crosslinking degree decreases [34,35], the studied hydrogels indicated the pore dimension increasing along the crosslinking degree increasing. Moreover, the increase in water content led to a less uniform distribution of the pore diameters, most probably due to the diminishing of the intermolecular physical connections amongst the chitosan chains and P5P.

### 3.3. The Hydrogel Stability over Time

The paper’s aim was to reach self-healing hydrogels that can be easily injected, and to this end, hydrogels with high water content were prepared. As the ^1^H-NMR spectra indicated an equilibrium of the imination reaction, it is desirable to know how the stability (and implicit properties) of the hydrogels is affected by this equilibrium. ^1^H-NMR spectra recorded over time (1, 7, 15, and 22 days) evidenced the progressive diminishing of the integrals of imine and aldehyde protons and the appearance of the enol proton (around 6.5 ppm) [23] for the hydrogels with high water content (those prepared with chitosan solutions 0.6% and 0.43%) (Figure 3a). This suggested that a too diluted system favored the shifting of imination to the reagents and the stabilization of the enol form of aldehyde. No such behavior was noticed for the hydrogels with lower water content (those prepared with 3% and 1% chitosan solutions); the hydrogel spectra remained unaffected over 22 days (Figure 3b). The ^1^H-NMR modification of the hydrogels with higher water content was accompanied by the gel–sol transition, pointing to their limited storage duration at room temperature, less than 22 days.

As the hydrogels were thought as SH hydrogels to be injected at targeted sites in the body, their stability/degradation behavior was investigated in media of different pH: 5.6, 6.8, 7.4, and 8, corresponding to different body fluids (tumors, normal tissue, peritoneal fluid) [36,37]. It is known that hydrogels swell in aqueous media due to their ability to absorb high water content, and their swelling rate is influenced by the pH and the hydrogel porosity. Concurrently, the pH triggers different dissolution patterns of the hydrogels [20,21,22,23,24,25,26,38,39,40]. In this context, the stability of hydrogels was monitored in media of four different pHs (Figure 4). No visual alteration of the hydrogels was seen over 9 days of investigation. Nevertheless, gravimetric monitoring revealed a progressive degradation, reaching a mass loss from 13 to 56%, function of media pH and hydrogel composition (Figure 4). As expected, the hydrogels degraded faster in acidic acetate buffer (pH = 5.6), in agreement with the shifting of imination equilibrium to the reagents and faster dissolution of chitosan under the influence of protons [41,42]. Interestingly, against the expectations, the mass loss decreased as the content of P5P decreased. A possible explanation for this can be the structural similarity of the phosphate group of the P5P unit with the phosphate buffer, which favored the dissolution.

### 3.4. Visual Testing of Self-Healing Ability of the Hydrogels

As was demonstrated in Section 3.1, the hydrogelation of chitosan with P5P was the result of the formation of reversible imine bonds and intermolecular physical interactions, both of them being favorable for the self-healing process [1,2,3,4,5,6]. The ability of hydrogels to self-heal was preliminarily investigated by the injection of the hydrogels through a syringe needle. As can be seen in Figure 5a–d,f, the hydrogels could easily pass through a needle of 0.6 mm without clogging and reshape the hydrogel. For better visualization of the self-healing, part of the hydrogel has been stained with methylene blue and injected next to a pristine hydrogel (Figure 5e,g), and they instantly connected, forming a single piece. This self-healing behavior points to an easy administration at specific sites, e.g., tumors or wounds.

### 3.5. Rheological Properties

As the studied hydrogels showed visual self-healing behavior, the next step to obtain more information on the recovery degree and to have an insight into their mechanical stability was to investigate their rheological properties [43,44]. The shear stress dependence of viscoelastic moduli, determined by amplitude sweep measurements, showed the gel-like behavior of the investigated samples (G′ > G″) (Figure 6). G′ and G″ values were constant below the limiting shear stress, τ_l_, which corresponds to a limiting strain, γ_l_. Viscoelastic moduli start to decrease above τ_l_ due to the modification of sample structure, and after critical shear stress or strain (where G′ = G″), the network structure was destructed, the sample acquiring liquid-like properties with G′ < G″.

The increase in water content and the decrease in P5P amount led to the shortening of the linear viscoelastic range (LVR) (Figure 6, Table 2). Thereby, a double water amount in hydrogels caused a diminishing of almost one order of magnitude of the interval in which G′ and G″ are constant, i.e., below 40.8 Pa for the sample **2d** and below 4.7 Pa for the sample **2q** (Figure 6a). The decrease in P5P amount in the starting system (higher NH_2_/CHO ratio) determined the decrease in the limit stress of LVR, e.g., from 40.8 Pa (sample **2d**) to 0.6 Pa (sample **4d**) (Figure 6b).

The effect of water content and NH_2_/CHO ratio on the G′ and G″ values was also investigated by frequency sweep measurements at a strain of 1% from LVR (Figure 7). G′ and G″ decreased when the water content in hydrogels decreased due to the reduced probability of chitosan-P5P-chitosan contacts, which leads to decreased crosslinking capacity (Figure 7a). Thus, a double amount of water inflicted the diminishing of the G′ and G″ values (at 1 rad s^−1^) from 326 Pa and 28.7 Pa (sample **2d**) to 20.2 Pa and 2.8 Pa (sample **2q**).

Generally, a lower amount of crosslinker causes the formation of a smaller number of bridges between the chitosan chains and a decrease in the density of the hydrogel network, leading to worsened viscoelastic properties. This general rule also applied to the studied hydrogels, with G′ and G″ reducing by two orders of magnitude from 326 Pa and 28.7 Pa for the sample **2d** (NH_2_/CHO = 2) to 3.4 Pa and 0.9 Pa for the sample **4d** (NH_2_/CHO = 4), confirming a higher crosslinking degree for the hydrogel samples prepared with higher P5P amounts (Figure 7b, Table 2).

Considering the rheological data obtained by the amplitude and frequency sweep tests, in Figure 8, it is illustrated the effect of water content and NH_2_/CHO ratio of the hydrogels on the γ_l_ and the loss tangent (tan δ = G″/G′). The loss tangent, tan δ, gives the information about the viscoelasticity degree of the material in the investigation conditions: tan δ > 1 is characteristic for liquid-like materials, and tan δ < 1 is characteristic for solid-like materials [45]. The widest ranges of linear viscoelasticity were recorded for the hydrogels with NH_2_/CHO ratios up to 3, with double water amount (Figure 8a). The other samples exhibited the limit of LVR below 10 Pa (Table 2). The samples with NH_2_/CHO ≤ 2.5, regardless of the water content, exhibit the lowest values of tan δ due to their stronger network (Figure 8b).

The structural recovery ability of the samples was studied at 10 rads^−1^ by applying three shear steps: 1% (300 s); 100% (300 s); 1% (360 s) [46]. Figure 9a shows the effect of the NH_2_/CHO ratio on the G′ recovery degree for the samples **2d**, **3d**, and **4d**.

The samples displayed G′ values that remain constant during the measurements at low strain amplitude, and by applying a strain of 100%, G′ immediately decreases. At high deformation amplitude, a decrease in G′ was observed in the first seconds of shear and the constant value was reached after about 150 s. Upon the removal of the high strain pulse, G′ quickly increases, recovering partially or totally the network structure. The structure recovery degrees of investigated samples show values higher than 74.7%, indicating a good thixotropic recovery (Figure 9b, Table 2) [47]. The increase in water content and decrease in the P5P content favored the structural recovery degree of the hydrogel network. The effect of water content on the structural recovery degree was more important for the samples containing a smaller amount of P5P crosslinker (higher NH_2_/CHO ratio). Thereby, for the sample with NH_2_/CHO = 2, the structural recovery degree increases from 74.7% for the sample **2d** to 80% for the sample **2q**. On the other hand, the structural recovery degree of the sample with NH_2_/CHO = 3 increased from 78.8% for **3d** to 94.7% for **3t**. The higher structural recovery degree of the samples with lower crosslinker amount and/or higher water content was correlated with the easier restoration of the physical forces in the less viscous hydrogel.

In conclusion, the rheological investigation of the effect of water content and crosslinker amount on the viscoelastic properties of the hydrogels shed light on their gel-like and thixotropic behavior. For all studied samples, it was evidenced the gel-like behavior with G′ greater than G″. The increase in the crosslinker content, on the one hand, widens the range of shear stability and enhances the mechanical properties and, on the other hand, decreases the structural recovery degree (thixotropic behavior). Analyzing both the effect of the amount of crosslinker and water content on the rheological properties, it was found that the optimum experimental conditions to obtain the convenient viscoelastic behavior correspond to NH_2_/CHO < 3, with a water content corresponding to the hydrogel preparation from chitosan solutions 1%.

Comparing the data with other SH chitosan hydrogels [11,12,13,14,15,16,17,48,49], it should be remarked that chitosan crosslinking with P5P yielded SH hydrogels for a large variety of compositions allowing the fine-tuning of the other properties in view of specific applications.

## 4. Conclusions

A large series of 30 hydrogels was prepared from chitosan and pyridoxal 5-phosphate, varying the glucosamine/aldehyde ratio from 1 to 9, and the water content by using chitosan solutions of 3, 1, 0.6, and 0.43%. The NMR, FTIR, X-ray, and POM investigation indicated that hydrogelation was guided by the formation of imine bonds and physical forces between chitosan chains and pyridoxal 5-phosphate. The hydrogels had microporous morphology with pore diameter controlled by the water content, degradability in aqueous media controlled by pH, and good stability over 22 days. By visual assessment, the studied hydrogels were capable of passing through a needle of 0.6 mm without clogging and reshaping the hydrogel, indicating self-healing ability. Deeper rheological investigations demonstrated that the viscoelastic properties decreased along with the increase in NH_2_/CHO ratio and water content, indicating the best mechanical properties for the hydrogels with NH_2_/CHO ≤ 3, regardless of the water content. On the other hand, the structural recovery degree increased when NH_2_/CHO and water volume in hydrogel increased, reaching values higher than 70% for the hydrogels with NH_2_/CHO ≤ 4. From the perspective of the application is injectable hydrogels, the best balance of mechanical properties/structure recovery degree was reached for the hydrogel prepared using an NH_2_/CHO ratio of 4 and chitosan solution of 1%. These results indicate the studied hydrogels as valuable vehicles for antitumor drug delivery.

## Data Availability

The data presented in this study are available on request from the corresponding authors.

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
