# Peer review of "Self-Healing Chitosan Hydrogels: Preparation and Rheological Characterization"

_polymers, 2022, doi:10.3390/polym14132570_

Round 1

Reviewer 1 Report

The study presented in the manuscript polymers-1766559, entitled “Self-healing chitosan hydrogels. Preparation and rheological characterization" by the authors Anda-Mihaela Craciun, Simona Morariu, and Luminita Marin, gives an interesting idea about chitosan crosslinked with 5-pyridoxal phosphate, with detailed experimental procedures followed. The reported results are very interesting and applicable for this study. I would like to advise authors for additional improvement of this paper quality by implementing minor corrections.

The abstract gives an overview of this study (goal, methods, results and conclusion), but, at the beginning, it is needed to emphasize the purpose of this research and give a broader context, especially for selected reagents (chitosan and 5-pyridoxal phosphate), with max 200 words.

Introduction provides detailed description of the state of the art, main aims and part of results. It is needed to explain why did authors chose the active form of vitamin B6 (5-pyridoxal phosphate) for self-healing hydrogels with chitosan?

Materials and Methods part provides data about used materials, equipment and applied methods for hydrogels synthesis, structural and supramolecular characterization and rheological properties. It is needed to add information on the purity of the used chemicals.

In the part Results and Discussion all experimental results are presented in details at 1 scheme, 9 figures and 2 tables. Scheme 1 should be larger and clearer. Figures 1b, 1c, 1d  should be clearer. "Figure 3" should be written in black color. Discussion of obtained results provides analysis of morphology, hydrogel stability over time, visual testing of self-healing ability of the hydrogels, and rheological properties. The authors should specify the criteria for the selection of representative samples from a series of 30 hydrogels. It is needed to explain why did authors chose the active form of vitamin B6 (5-pyridoxal phosphate) for self-healing hydrogels with chitosan? Could the self-healing hydrogel be appropriate for modified release of P5P after degradation? Why is 5-pyridoxal phosphate suitable for the delivery of antitumor drugs?

The Conclusions are very good and the scientific contribution is visible and applicable. The authors proposed the most optimal hydrogel sample, based on obtained results and pointed out future directions of research.

In this manuscript, authors cited 47 articles with relevant 10 auto citations.

It is needed to change template of this manuscript according Instruction and Polymers Journal`s template.

Please, it is needed to provide the full names of acronyms on first appearance in the abstract, manuscript and in figure captions (e.g. P5P, POM, NMR, FTIR, X-ray, SEM). The abbreviation P5P should be written in regular style without bold.

Please, it is needed to avoid 1st person plural and rewrite all sentences in 3rd person plural and passive voice - page 2 paragraph 2.

Please be consistent with the style of references.

I recommend the acceptance of the manuscript in the journal Polymers with minor corrections on above mentioned suggestions and comments.

Best regards,

Reviewer

Author Response

Answer to reviewers

First of all, we want to thank to reviewers for their effort to read and to do proper observations on our manuscript. Your expertise and time are highly appreciated.

We carefully considered all the comments and amended the manuscript, hoping that now it was improved for the reader’s benefit. All the changes in the text were highlighted in red letters. The answer to the comments is as follows.

Reviewer 1

Reviewer comment:

The study presented in the manuscript polymers-1766559, entitled “Self-healing chitosan hydrogels. Preparation and rheological characterization" by the authors Anda-Mihaela Craciun, Simona Morariu, and Luminita Marin, gives an interesting idea about chitosan crosslinked with pyridoxal 5-phosphate, with detailed experimental procedures followed. The reported results are very interesting and applicable for this study. I would like to advise authors for additional improvement of this paper quality by implementing minor corrections.

The abstract gives an overview of this study (goal, methods, results and conclusion), but, at the beginning, it is needed to emphasize the purpose of this research and give a broader context, especially for selected reagents (chitosan and pyridoxal 5-phosphate), with max 200 words.

Authors answer:

The Abstract has been modified to highlight the purpose of the research, which was the preparation of chitosan self-healing hydrogels as carriers for local drug delivery through parenteral administration.    

Reviewer comment:

Introduction provides detailed description of the state of the art, main aims and part of results. It is needed to explain why did authors chose the active form of vitamin B6 (pyridoxal 5-phosphate) for self-healing hydrogels with chitosan?

Authors answer:

The paper aim was to evidence self-healing hydrogels as carriers for local drug delivery by parenteral administration, the anticancer drugs being especially envisaged. With this aim in mind, pyridoxal 5-phosphate has been used as chitosan crosslinker, due to its benign nature (it is a food supplement approved by FDA) and its preventive tumorigenesis activity and synergistic effect with some antitumor drugs, which create the premises for active multifunctional hydrogel carriers in anticancer drug delivery. This was better highlighted in Abstract and Introduction. 

Reviewer comment:

Materials and Methods part provides data about used materials, equipment and applied methods for hydrogels synthesis, structural and supramolecular characterization and rheological properties. It is needed to add information on the purity of the used chemicals.

Authors answer:

The purity of materials has been provided in the experimental section.

Reviewer comment:

In the part Results and Discussion all experimental results are presented in details at 1 scheme, 9 figures and 2 tables. Scheme 1 should be larger and clearer. Figures 1b, 1c, 1d  should be clearer. "Figure 3" should be written in black color. Discussion of obtained results provides analysis of morphology, hydrogel stability over time, visual testing of self-healing ability of the hydrogels, and rheological properties. The authors should specify the criteria for the selection of representative samples from a series of 30 hydrogels. It is needed to explain why did authors chose the active form of vitamin B6 (pyridoxal 5-phosphate) for self-healing hydrogels with chitosan? Could the self-healing hydrogel be appropriate for modified release of P5P after degradation? Why is pyridoxal 5-phosphate suitable for the delivery of antitumor drugs?

Authors answer:

Scheme 1 has been improved for clarity.

Figures 1b,c,d were replaced with clearer ones.

“Figure 3” was written in black colour.

The figures given as representative in the Results and Discussion part, were representative for the behaviour discussed, usually containing different glucosamine/aldehyde ratio, or different water content. This was better highlighted along the paper.

As highlighted in Introduction, P5P demonstrated synergistic effect with some antitumor drugs. In this light, it is expected as its release during the hydrogel biodegradation to enhance the antitumor drug activity. Regarding the activity of the modified form of P5P (the enol form), it should be highlighted that P5P exists in these two tautomeric forms, no activity differences being reported.

Reviewer comment:

The Conclusions are very good and the scientific contribution is visible and applicable. The authors proposed the most optimal hydrogel sample, based on obtained results and pointed out future directions of research.

In this manuscript, authors cited 47 articles with relevant 10 auto citations.

It is needed to change template of this manuscript according Instruction and Polymers Journal`s template.

Please, it is needed to provide the full names of acronyms on first appearance in the abstract, manuscript and in figure captions (e.g. P5P, POM, NMR, FTIR, X-ray, SEM). The abbreviation P5P should be written in regular style without bold.

Please, it is needed to avoid 1st person plural and rewrite all sentences in 3rd person plural and passive voice - page 2 paragraph 2.

Please be consistent with the style of references.

I recommend the acceptance of the manuscript in the journal Polymers with minor corrections on above mentioned suggestions and comments.

Authors answer:

The template of the manuscript has been changed according to Polymers template.

The full name of the acronyms was given on the first appearance in manuscript. The abbreviation of P5P was written in regular style.

Along the paper the 1st person plural was avoided. Nevertheless, in the paragraph 2 (page 2) we prefer to let 1st person plural, as this crosslinking pathway of chitosan with monoaldehydes has been evidenced and developed in our group. We are seeking for reviewer opinion regarding this.

The references were reviewed for consistency.

Reviewer 2 Report

The paper reports preparation self-healing hydrogels based on chitosan and 5-pyridoxal phosphate, the active form of vitamin B6. These results indicate the studied hydrogels as interessing vehicles for antitumor drug delivery. The work is very interesting, well structured and we recommend its acceptance for publication.

Author Response

Answer to reviewers

First of all, we want to thank to reviewers for their effort to read and to do proper observations on our manuscript. Your expertise and time are highly appreciated.

We carefully considered all the comments and amended the manuscript, hoping that now it was improved for the reader’s benefit. All the changes in the text were highlighted in red letters. The answer to the comments is as follows.

Reviewer 2:

Reviewer comment:

The paper reports preparation self-healing hydrogels based on chitosan and 5-pyridoxal phosphate, the active form of vitamin B6. These results indicate the studied hydrogels as interesting vehicles for antitumor drug delivery. The work is very interesting, well structured and we recommend its acceptance for publication.

Authors answer:

Thank you so much for your positive appreciation of our work.

Reviewer 3 Report

Presented article shows the results of self-healing hydrogels based on chitosan and 5-pyridoxal phosphate in different ratio and with different content of water. Investigated hydrogels were characterized by NMR, FTIR, X-ray and POM measurements. Additionally, Degradability of the hydrogels was studied in media of four different pH, and preliminary self-healing ability was visual established by injection through a syringe needle. The presented results are interesting and worth to publish after minor revisions.

The remarks and comments are the following:

3.2 Morphology

Please, provide the average pore size for each hydrogel type. Please add also the graph presented the relationship between pore size and water content of hydrogels..

Graphs

Please, add the error bars.

Hydrophilicity of hydrogels

In order to check the hydrophilicity of investigated hydrogels, the degree of swelling is measured. Please do these measurement.

Comparison with another membranes

Please, compare obtained hydrogels with these reported in literature.

References

Please, correct the listed references:

[7] DOI is missing; please remove the dot at the end

[16] please remove the dot at the end

[19] DOI is missing

[21] DOI is missing; please remove semicolon

[28], [34] please, replace the comma with a dot before the DOI

[42]  DOI is missing

[46] please, write DOI without emphasis

Author Response

Answer to reviewers

First of all, we want to thank to reviewers for their effort to read and to do proper observations on our manuscript. Your expertise and time are highly appreciated.

We carefully considered all the comments and amended the manuscript, hoping that now it was improved for the reader’s benefit. All the changes in the text were highlighted in red letters. The answer to the comments is as follows.

Reviewer 3

Reviewer comment:

Presented article shows the results of self-healing hydrogels based on chitosan and pyridoxal-5- phosphate in different ratio and with different content of water. Investigated hydrogels were characterized by NMR, FTIR, X-ray and POM measurements. Additionally, Degradability of the hydrogels was studied in media of four different pH, and preliminary self-healing ability was visual established by injection through a syringe needle. The presented results are interesting and worth to publish after minor revisions.

The remarks and comments are the following:

3.2 Morphology

Please, provide the average pore size for each hydrogel type. Please add also the graph presented the relationship between pore size and water content of hydrogels.

Authors answer:

Histograms on the SEM images were done and included in the Figure 2.

Reviewer comment:

Graphs

Please, add the error bars.

Authors answer:

Error bars were included. However, in some cases, the error bars were too small and could not be observed. In order to clarify this aspect, the following sentence was included in the section 1.3 Equipment and methods: “The rheological measurements were performed at 37°C in duplicate to estimate the results variability, using fresh sample for each test. The errors were lower than 13%. For some strongly crosslinked or undiluted samples, no rheological measurements could be performed due to their brittle properties.”

Reviewer comment:

Hydrophilicity of hydrogels

In order to check the hydrophilicity of investigated hydrogels, the degree of swelling is measured. Please do these measurement.

Authors answer:

We agree with reviewer that swelling degree and hydrophilicity are important parameters for hydrogels. Nevertheless, in the due time allocated by the journal for manuscript revision, we could not perform such studies. Considering that the focus of the manuscript was the self-healing behaviour of the hydrogels, we believe that the lack of swelling studies will not affect the paper meaning. We are seeking for the reviewer opinion regarding this.

Reviewer comment:

Comparison with another membranes

Please, compare obtained hydrogels with these reported in literature.

Authors answer:

A comparison with other SH chitosan hydrogels reported in literature it is difficult to do, because different parameters were monitored and reported. However, compared to other hydrogels reported in literature, the hydrogels reported in this manuscript present the advantage of a large variety of compositions for which the self-healing behaviour was reached, fact which allow to tune the other properties in view of specific application. Changes in the manuscript were done to highlight this and two new references were given (references 48, 49).

Reviewer comment:

References

Please, correct the listed references:

[7] DOI is missing; please remove the dot at the end

[16] please remove the dot at the end

[19] DOI is missing

[21] DOI is missing; please remove semicolon

[28], [34] please, replace the comma with a dot before the DOI

[42]  DOI is missing

[46] please, write DOI without emphasis

Authors answer:

The references style has been corrected.